# Using Species Distribution Models (SDMs) to Estimate the Suitability of European Mediterranean Non-Native Area for the Establishment of *Toumeyella Parvicornis* (Hemiptera: Coccidae)

**DOI:** 10.3390/insects14010046

**Published:** 2023-01-03

**Authors:** Nicolò Di Sora, Roberto Mannu, Luca Rossini, Mario Contarini, Diego Gallego, Stefano Speranza

**Affiliations:** 1Dipartimento di Scienze Agrarie e Forestali, Università degli Studi della Tuscia, Via San Camillo de Lellis snc, 01100 Viterbo, Italy; 2Dipartimento di Agraria, Università degli Studi di Sassari, Viale Italia 39A, 07100 Sassari, Italy; 3Service d’Automatique et d’Analyse des Systèmes, Université Libre de Bruxelles, v. F.D. Roosvelt 50, CP 165/55, 1050 Brussels, Belgium; 4Department of Ecology, University of Alicante, Carretera San Vicente del Raspeig s/n, 03690 San Vicente del Raspeig, Alicante, Spain

**Keywords:** pine tortoise scale, alien species, stone pine, species distribution models, biological invasion

## Abstract

**Simple Summary:**

Predicting species distribution is a fundamental step for setting up opportune control actions. The suitability of the environment for the establishment of the species is even more important in case of invasive insects, such as *Toumeyella parvicornis* (Hemiptera, Coccidae). This species is a soft scale insect native to North America recently introduced in Italy and in France, where it established and spread, causing harmful infestations on stone pine (*Pinus pinea* L.) plants. Some aspects of its biology, such as the several overlapped generations and the high fecundity, may contribute to make it a successful invasive species when in a suitable environment. This is supported by the observations carried out in Central Italy, where the climate and environmental conditions seem suitable for the species to develop. To prevent further spread across the Mediterranean basin, it would be helpful to identify the most suitable areas by considering bioclimatic variables, as is commonly carried out in case of invasive species. We prepared potential pest distribution maps of European areas by utilizing Species Distribution Models. This information adds further detail to the report recently published by the European Food Safety Authority (EFSA). The areas with the highest suitability for the species are located along the coasts, where most Mediterranean pines occur. This correspondence suggests a high risk of widespread dispersal and provides useful information for implementing management strategies of this damaging pest.

**Abstract:**

The pine tortoise scale, *Toumeyella parvicornis*, is an insect native to the Nearctic region that is able to infest several *Pinus* species. It can cause weakening, defoliation and, at high infestation levels, tree death. After its first report in Italy in 2015, the pest spread rapidly over the surrounding areas and was reported in France in 2021. Due to the threat that this pest poses to pine trees, the suitability of European Mediterranean basin areas for *T. parvicornis* at different spatial scales was estimated by constructing species distribution models (SDMs) using bioclimatic variables. Our results showed that several coastal areas of the Mediterranean basin area could be suitable for *T. parvicornis*. Based on performance assessment, all the SDMs tested provided a good representation of the suitability of European Mediterranean non-native area for *T. parvicornis* at different spatial scales. In particular, most of the areas with a medium or high level of suitability corresponded to the geographical range of distribution of different *Pinus* spp. in Europe. Predicting the suitability of European Mediterranean areas for *T. parvicornis* provides a fundamental tool for early detection and management of the spread of this pest in Europe.

## 1. Introduction

The pine tortoise scale, *Toumeyella parvicornis* (Cockerell) (Hemiptera: Coccidae), is a sap-sucking insect infesting several species belonging to *Pinus* genera (Pinaceae) [1,2]. The insect mainly lives on the crown [1], and adult females (Figure 1) tend to colonize more small twigs than needles [3,4]. *Toumeyella parvicornis* generally overwinters as fertilized adult females [5] and are able to complete at least 1–2 generations per year in its native range [1]. Nymphs develop to the adult stages through three instars in female individuals and four in males. Adults exhibit a marked sexual dimorphism with wingless females. Except for the first preimaginal stage, usually defined as “crawler”, *T. parvicornis* is sessile [2,6].

*Toumeyella parvicornis* originates from the Nearctic region, where it was first described in Florida [7] and then reported in the northern part of the USA [7,8,9,10], in Canada [11], and in Mexico [12]. In recent years, *T. parvicornis* has been established in other countries outside its native area, including the Turks and Caicos Islands [2], Puerto Rico [13], and Europe [14]. The first report of this insect in Europe was in Italy [14], where the first outbreak was documented on *Pinus pinea* L. trees in the Campania region. Recently, *T. parvicornis* has rapidly spread throughout Italy [4,15], and in 2021, it was reported in France [15].

The rapid expansion of the pine tortoise scale in Europe is generating serious concerns throughout the scientific community. In fact, *T. parvicornis* attacks cause substantial weakening and defoliation of pine trees that can lead to tree death when severe infestations occur [16]. *Toumeyella parvicornis* populations, which may rapidly reach a high density, increase the risk of accidental branch falls, representing a great concern for citizens’ safety and health in urban green areas. In addition, *T. parvicornis* produces a large amount of honeydew, which can disturb urban park users especially in urban environments [16], where pine trees often have a fundamental historical and architectural value [4,17,18], since it covers structures and monuments. The implementation of monitoring strategies is of fundamental importance mainly because pine trees are very common in urban areas, where control actions can be strongly limited by European laws even when pest infestations can endanger the safety of citizens [19].

Although *Pinus banksiana* Lamb represents the main host of *T. parvicornis* in the Nearctic region [11,20], this insect can develop on different *Pinus* species in non-native areas. For instance, in the Turks and Caicos Islands, the Caribbean pine [*Pinus caribaea* var. *bahamensis* (Grisebach) W. H. Barrett & Golfari] is the primary host of this pest [2], whereas in Europe the stone pine (*Pinus pinea* L.) is its main host [6]. *Pinus pinaster* Aiton showed low susceptibility to pine tortoise scale, whereas *Pinus halepensis* Mill. was apparently not susceptible to its infestations [6].

Among different statistical and mathematical tools, Species Distribution Models (SDMs) represent the most widely used tool in the research field of ecology and conservation biology to assess the potential distribution of a species. In particular, SDMs facilitate our understanding of species-environment relationships and estimate species potential distribution also in areas where the species has not been observed yet [21,22,23].

Although SDMs do not report any information about the temporal dynamics of the diffusion of a species [24], they are widely used to make projections about the suitability of different geographical environments to invasive alien species [25,26,27,28,29], also in the case of small sample size [30].

Identifying the most suitable areas for an invasive species could actively support local authorities, technicians, and communities in defining the areas at risk for invasion and where surveillance should be improved. For this reason, the aim of this work was to estimate the potential suitability of European Mediterranean non-native areas for *T. parvicornis*. In particular, different SDM approaches were used to identify the potential spread of the pine tortoise scale in Italy and Europe on the basis of known distribution and bioclimatic variables. The comparison between different SDMs aimed to provide a better scenario of the potential distribution of this species in European Mediterranean areas.

## 2. Materials and Methods

### 2.1. Occurrence Records and Bioclimatic Variables

Geographic coordinates for *T. parvicornis* occurrences in Mediterranean Europe were obtained from literature and in Central Italy by direct monitoring in stone pine growing-areas. In particular, reports from the Regional Plant Protection Services of the Campania region [available at http://agricoltura.regione.campania.it/ (accessed on 1 December 2022)] and the European and Mediterranean Plant Protection Organization (EPPO) database [15] were used to extract the occurrences of pine tortoise scale in Central Italy and Southern France, respectively. In Central Italy, records of *T. parvicornis* were also acquired by monitoring stone pine growing-areas and trees in urban parks of the Lazio region from December 2021 to January 2022. At first, sampling sites representing *P. pinea* growing-areas and urban trees were randomly chosen from a distribution map available for the Lazio region [31], and from the Google Earth software [available from https://earth.google.com/web/ (accessed on 1 December 2022)]. At each selected location, *T. parvicornis* infestations were assessed by visually inspecting the trees for typical symptoms (i.e., presence of black mold on the crown, plant deterioration, yellowish and desiccated needles as shown in Figure 2). A monocular scope at 40× magnification was used to detect the insect on symptomatic plants, whereas needles and branches on the ground were examined to assess the presence of *T. parvicornis* adults and/or molting residuals. A total of 52 and 53 occurrence records for *T. parvicornis* were found in Italy and Europe, respectively (Appendix A).

### 2.2. Bioclimatic Variables

Nineteen bioclimatic variables were obtained from the WorldClim Global Climate Database (version 1.4) [http://www.worldclim.org (accessed on 1 December 2022)] as georeferenced raster files [32]. Georeferenced images with a spatial resolution of 30 arc-sec (∼1 km^2^) and 2.5 arc-min (∼4.5 km^2^) were used for modelling *T. parvicornis* potential distribution in Italy and Europe, respectively. Prior to the analysis, raster images were clipped to cover the same fixed area using QGIS software (version 3.24) [available at http://qgis.org (accessed on 1 December 2022)]. Multicollinearity among the bioclimatic variables was tested at a spatial resolution of 30 arc-sec (∼1 km^2^) using the Pearson’s correlation coefficients in order to remove the highly correlated variables (R^2^ > 0.80 or R^2^ < −0.80) before running the models (Appendix A). The multicollinearity tests led us to select only nine bioclimatic variables (Table 1).

### 2.3. Species Distribution Modelling

Four different algorithms were used to estimate the potential distribution of *T. parvicornis* in Italy and Europe: Generalized Linear Model (GLM), Multivariate Adaptive Regression Splines (MARS), Random Forest (RF), and MaxEnt [33,34]. The models were implemented using the biomod2 package [35] in R software (version 4.1.3) [36].

The GLM, MARS and RF algorithms need pseudo-absence records in addition to occurrence points. Therefore, 1000 pseudo-absence records, located at least 500 m apart from the occurrence points, were generated. This process was repeated 3 times, thus obtaining three different datasets of pseudo-absences for each algorithm. The default model settings in biomod2 were used for fitting GLM, MARS, and RF algorithms.

The MaxEnt algorithm can be applied considering presence-only records instead of presence-absence data [37], which implies the use of complex nonlinear functions [38]. Model complexity can be controlled by a set of parameters named Feature Classes (FCs) and Regularization Multiplier (RM). The FCs aim to improve model fitting [38] and consist of a transformation of the original predictor variables that can be used either separately or in combination [i.e., linear (L), quadratic (Q), hinge (H), product (P), and threshold (T)], whereas the RM aims to reduce overfitting [39].

Parameterization in MaxEnt is necessary as models fitted with default settings often lead to skewed projections if compared to the parametrized models [39,40,41,42]. For this reason, twenty different models were fitted by considering 4 FC combinations (L, LQ, LQH, H) and RM values ranging from 1 to 5. All the recorded occurrences available and 1000 randomly positioned background points were used for model fitting. The model best fitting the data was selected based on the Akaike’s Information Criterion with a small sample size correction (AICc) [43], which reflects both model goodness-of-fit and complexity [44]. Conventionally, the lowest AICc value indicates the best fitting model for the dataset considered [44,45]. Parametrization was carried out using the ENMeval package [45] in R software.

### 2.4. Model Evaluation and Predictions

For all the tested models, 80% of the data was used for calibration, whereas the remaining 20% was used for performance tests, as conventionally applied for SDM application and modelling [29,46,47].

Model performance was evaluated by considering the Area Under the Curve (AUC), the Receiver Operating Characteristic (ROC), and the True Statistical Skill (TSS). In short, AUC measures the discrimination capacity between presences and absences and ranges from 0 to 1. Values of AUC higher than 0.8 and 0.9 indicate good and excellent performances, respectively [48,49,50,51]. The TSS measures the classification accuracy of a model with respect to a threshold value and ranges between −1 and 1. TCC values of −1 and 1 indicate no correspondence and complete correspondence between observations and predictions, respectively [52].

Finally, the distribution probabilities obtained by the models were projected using QGIS software to visually evaluate the suitability level of European countries for *T. parvicornis* development. The variables with the greatest contributions to the distribution of *T. parvicornis* were conventionally divided in “classes of suitability” that ranged from “very low” to “very high” suitability (S), as follows: S < 0.2 = very low, 0.2 ≤ S < 0.4 = low, 0.4 ≤ S < 0.6 = intermediate, 0.6 ≤ S < 0.8 = high, and S ≥ 0.8 = very high.

## 3. Results

### 3.1. Habitat Suitability for T. parvicornis in Italy

The results of the contribution weights of bioclimatic variables in estimating the suitability of Italian areas for *T. parvicornis* varied with the model involved. The bioclimatic variables influencing the most the distribution of *T. parvicornis* in Italy were BIO01 (i.e., annual mean temperature) and BIO12 (i.e., annual precipitation) for the GLM model, BIO01 and BIO03 (i.e., isothermality) for the MARS model, BIO03 and BIO01 for the RF model, and BIO01 and BIO15 (i.e., precipitation seasonality) for the MaxEnt model (Table 2).

Assessment of SDMs performance when determining the areas suitable to *T. parvicornis* in Italy was based on the AUC and TSS values. All models provided excellent performances considering that AUC values were higher than 0.93, indicating a strong capability to predict the potential distribution of *T. parvicornis* in Italy. However, the RF model (AUC = 0.99 ± 0.01; TSS = 0.96 ± 0.04) had the best performance compared to the other tested models (MARS: AUC = 0.97 ± 0.02, TSS = 0.93 ± 0.05; MaxEnt: AUC = 0.96 ± 0.05, TSS = 0.90 ± 0.10; GLM: AUC = 0.96 ± 0.07, TSS = 0.90 ± 0.10).

Model predictions for the Italian Peninsula are shown in Figure 3. Despite the observed differences, both in the weight of bioclimatic variables and in predictive power, no specific differences in prediction of the most suitable area for *T. parvicornis* were identified among the tested SDMs. The potential *T. parvicornis* distribution area was mainly concentrated around the coastal areas of Lazio, Campania, and Toscana regions (Figure 3). Moreover, high suitability areas were found in the Sardinia region by GLM, MARS, and MaxEnt models, whereas only the GLM model predicted a high suitability area in the middle of Puglia and Calabria regions.

Larger differences among the model predictions concerned the lower-suitability areas (Figure 3) where the MARS model extended the potential distribution more than the other models. On the contrary, the potential distribution estimated by the RF model was quite restricted.

### 3.2. Habitat Suitability for T. parvicornis in European Mediterranean Area

As already noted for Italy, the contribution of bioclimatic variables in estimating the most suitable areas for *T. parvicornis* in European Mediterranean area was different among the models tested. The BIO01 (i.e., annual mean temperature) and BIO19 (i.e., precipitation of the coldest quarter) variables were the most important variables affecting the distribution of *T. parvicornis* for the GLM and MARS models (Table 3). The bioclimatic variables contributing the most to the projections in the MaxEnt model were BIO15 (i.e., precipitation seasonality) and BIO19, whereas those contributing the most in the RF model were BIO01 and BIO19. Among all the bioclimatic variables considered, BIO02 (i.e., mean diurnal temperature range) and BIO04 (i.e., temperature seasonality) showed limited contribution to estimate the potential species distribution (Table 3).

Based on AUC and TSS, the MARS model best represented the dataset available (AUC = 0.98 ± 0.03; TSS = 0.96 ± 0.06), followed by GLM (AUC = 0.97 ± 0.04; TSS = 0.94 ± 0.07), MaxEnt (AUC = 0.97 ± 0.03; TSS = 0.94 ± 0.07), and RF (AUC = 0.97 ± 0.03; TSS = 0.94 ± 0.07), which showed very similar values.

A visual description of the results is provided in Figure 4. As already observed in the Italian projections, the most suitable areas to *T. parvicornis* in European Mediterranean area were concentrated in the coastal zone (Figure 2). Considering the overall model predictions, the most suitable areas were the Tyrrhenian Sea coast of Italy, the French Riviera, and the Balkan coast of the Adriatic Sea. In addition, a relevant extension of the potential distribution predicted by the models was in the north-western area of the Iberian Peninsula covering both Portugal and Spain. Unlike the other models, GLM and MARS indicated highly suitable areas for *T. parvicornis* along the e astern coast of Spain as well.

## 4. Discussion

The recent detection of *T. parvicornis* in Italy and in France has raised a serious concern about the potential diffusion of this pest across the European continent, including the Mediterranean area. Our study estimated for the first time the most suitable areas where the species may potentially spread in the next few years using a species distribution modelling approach. SDMs are a powerful and widely used tool to predict the areas where a species may potentially develop and spread [29,53,54,55,56,57]. All the tested models demonstrated good accuracies to estimating the suitability of Mediterranean European countries for *T. parvicornis*, and variability in accuracies were exclusively due to differences in fitting functions between regression-based (e.g., GLM, MARS) and nonparametric models (e.g., RF) [42].

Based on our study, the annual mean temperature played a fundamental role in explaining the projections of habitat suitability for *T. parvicornis* in Italy, especially in the GLM, MARS, and MaxEnt models. This information is of fundamental importance in the framework of a climate change scenario, as already stated for other pest species infesting agricultural and forest environments, such as *Philaenus spumarius* L. (Hemiptera: Aphrophoridae) [57], *Lobesia botrana* (Denis & Schiffermüller) (Lepidoptera: Tortricidae) [58], and *Tuta absoluta* (Meyrick) (Lepidoptera: Gelechiidae) [59]. Our results are also in line with the findings of Solhjouy-Fard et al. [60] and Yan et al. [61], where the annual mean temperature was the variable best explaining the potential distribution of *Ferrisia virgata* (Cockerell) (Hemiptera: Pseudococcidae), *Bemisia tabaci* (Gennadius) (Hemiptera: Aleyrodidae), *Apodiphus amygdali* (Germar) (Hemiptera: Pentatomidae), *Adelphocoris lineolatus* (Goeze) (Hemiptera: Miridae), and *Thrips palmi* Karny (Thysanoptera: Thripidae).

A secondary variable that is worth mentioning for Italy is isothermality, which was the best explanatory variable according to the RF model. This variable is strictly related to the maritime areas, given that it evaluates the thermal gap between summer and winter temperatures [62]. Areas characterized by warmer winters, which Italian maritime areas normally have, may favor the development of *T. parvicornis* by leading this species to have more generations than in places with colder winters. In fact, this phenomenon has been already observed for other alien species, such as *Nezara viridula* (Linnaeus) (Hemiptera: Pentatomidae) and *Hyphantria cunea* (Drury) (Lepidoptera: Erebidae) [63,64,65,66,67], and may also occur with *T. parvicornis*.

The European scenario outputs provided more extensive projections than the ones obtained that only considered Italy. In fact, the GLM and MARS models confirmed that temperature was the main explanatory variable at European Mediterranean spatial scale, whereas MaxEnt and RF considered precipitation seasonality and precipitation of the coldest quarter, respectively, as the best ones. Precipitation seasonality is a relevant variable, above all, in the coastal areas featured by rainfall anomalies during the seasons [62] and where the pest is already present (e.g., France). However, strong precipitation during summer, when the *T. parvicornis* activity is higher, may act as a natural control of preimaginal stages (i.e., the crawler). In fact, intense rainfall can kill the young instars and the mobile stages of this insect or carry them away from the host plant [16], as already observed for other species such as *Dactylopius opuntiae* (Cockerell) (Hemiptera: Dactylopiidae), *Quadricalcarifera punctatella* (Motschulsky) (Lepidoptera: Notodontidae), and *Eriogaster lanestris* (Linnaeus) (Lepidoptera: Lasiocampidae) [68,69,70]. In contrast, precipitation of the coldest quarter is a variable that may affect the overwintering capacity of the pest. In fact, strong rainfall in winter may mechanically remove the overwintering individuals from the host plant, thus influencing de facto the abundance of the population of the subsequent spring [71].

Our findings are also in line with Bragard et al. [5], who found that European countries have suitable features for the development of this pest, using the World distribution of Köppen–Geiger climate types present in the areas where *T. parvicornis* occurred as a method to predict the potential distribution of the insect.

The main host species of *T. parvicornis* occurring in Europe is *P. pinea*. This plant species has a low genetic diversity [72,73,74], which may favor the spread of this insect. In fact, a lower genetic diversity may indicate similar susceptibility among the plants to the pest, above all where extensive monocultures of *Pinus* species are present. This aspect has been already observed in intensive monoculture fields where a low genetic diversity of the plants resulted in a higher susceptibility to both biotic and abiotic stresses [75,76,77].

Stone and maritime pines are an essential element of cities in Italy [78] and coastal landscapes of many other Mediterranean countries such as France. Therefore, *T. parvicornis* may potentially change the aesthetic of many places if no prompt actions are carried out. Aesthetics is not the only element that may be impaired by the pest activity because *T. parvicornis* infestations may represent a serious concern where stone pines are cultivated for pinenut production, especially in some widely extended areas of Spain, Portugal, and France. Therefore, the interpretation of SDMs’ projections cannot ignore the host plant distribution. From this point of view, our results are in line with the suggestion provided by the European Food Safety Authority [5], indicating that *T. parvicornis* has the potential to establish throughout the Europe wherever suitable hosts occur. In fact, the most suitable areas for *T. parvicornis* in European Mediterranean countries estimated by SDMs overlapped with the distribution of *P. pinea* and *P. pinaster* (Figure 5), which were recognized as the main hosts in newly invaded areas in Europe [6].

In this regard, based on the information available on the distribution of the main host plant species in Italy and Mediterranean Europe, we analyzed where the potential spread of *T. parvicornis* is more likely to occur, among the areas indicated by the SDMs outputs of our study. In Mediterranean countries, most *Pinus* species are in coastal areas [79], where the model outputs suggested a high suitability for the pest diffusion. In the Italian Peninsula, the most infested areas cover the Lazio and Campania regions, where stone pines grow in both natural and urban environments. In addition, first detections have been recently reported also along the Adriatic coast, particularly in the Abruzzo and Puglia regions [15,80]. According to the model outputs, particular attention should be given to the extensive stone pine forests of Emilia Romagna, where *T. parvicornis* may aggravate the ongoing deleterious activity of *Crisicoccus pini* (Kuwana) (Hemiptera: Pseudococcidae), another invasive pest infesting the same host plant [81,82]. Similarly, Tuscany is another Italian region extensively covered with stone pines where the pest has been recently detected [83] and where the model outputs indicated a high suitability for the pest diffusion. Hence, all these Italian regions should be strictly monitored to promptly reduce the diffusion of *T. parvicornis* given the extensive presence of host plants and the high suitability indicated by our results.

A different scenario concerns Sardinia, where *T. parvicornis* has not been detected yet but stone and maritime pines are widely diffused [84,85], above all in the southern part of the region. In this case, careful monitoring of imported pine plants and wood is fundamental to prevent the introduction of this pest in the island.

Although *T. parvicornis* is an oligophagous species strictly related to a restricted pool of host plants, it has several characteristics of an ideal invasive species. In fact, small size, parthenogenetic reproduction, and high adaptability to new environments are features generally associated with highly invasive pests [86]. Moreover, the reduced size (up to 5 mm for adults) of *T. parvicornis* makes the detection of the infested plants, especially adult plants, very difficult and promotes its diffusion by wind. In addition to being small, the Italian and European climate conditions seem to play a fundamental role in its diffusion. For instance, in North America *T. parvicornis* has one generation per year [1,87], whereas in warmer areas several overlapped generations were observed [2,6].

Considering the distribution of both *P. pinea* and *P. pinaster* in Europe and our SDMs outputs, we hypothesize that the Iberian Peninsula may be another highly suitable area to *T. parvicornis*, given the extensive presence of the host plant [88,89]. Extensive pine forests and plantations are present in Portugal and Spain [79], where they are considered a precious source of biodiversity [90]. However, pine forests of the Iberian Peninsula are often subjected to abiotic stresses such as fires [91,92], and biotic adversities, such as the pine wood nematode *Bursaphelenchus xylophilus* Nickle (1981) [93,94] or bark beetle species belonging to the *Tomicus* genus [95,96]. Even though it has not been proven yet, it is likely that *T. parvicornis* could be more aggressive towards plants already affected by other adversities.

Obtaining a complete and detailed potential distribution map of *T. parvicornis* for different regions would be of fundamental importance, above all to construct prompt monitoring and control strategies. An early control strategy would avoid some dangerous conditions, as already observed in the Turks and Caicos Islands, where *T. parvicornis* dramatically reduced the range of native *Pinus caribaea* var. *bahamensis* [2]. Besides the short-term actions that could be performed based on the results of this study, a secondary aspect would be relevant for a long-term scenario. Therefore, we hypothesize that a slight increase of the annual mean temperature may dramatically extend the suitable areas. In fact, our predictions were based on actual climate conditions (years 1970–2000) because we aimed to understand the potential spread of the pest in the next few years. The same analysis presented in our work could be performed in further studies by considering a climate change scenario, as already carried out for other insect pest species [97,98].

In conclusion, we believe that the scenario presented in this work could allow technicians and local authorities to focus on the most susceptible areas to *T. parvicornis*, thus helping them to manage properly this injurious pest in Mediterranean European area.

## Figures and Tables

**Figure 1 insects-14-00046-f001:**
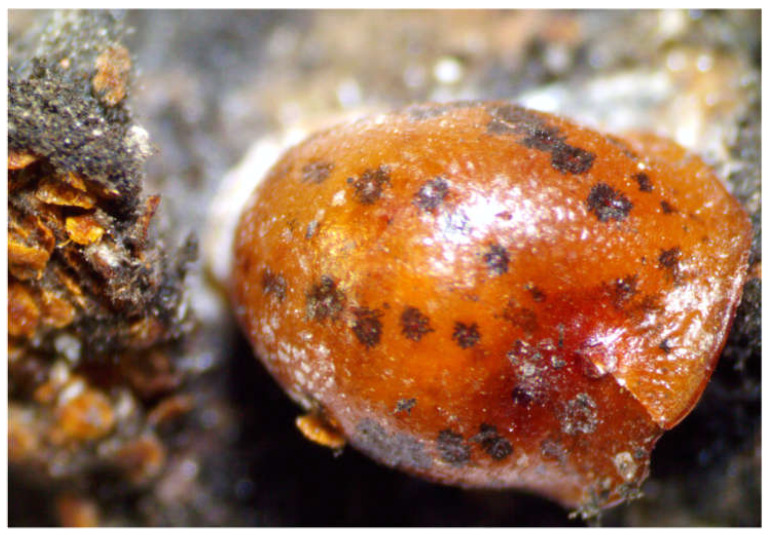
*Toumeyella parvicornis* adult female.

**Figure 2 insects-14-00046-f002:**
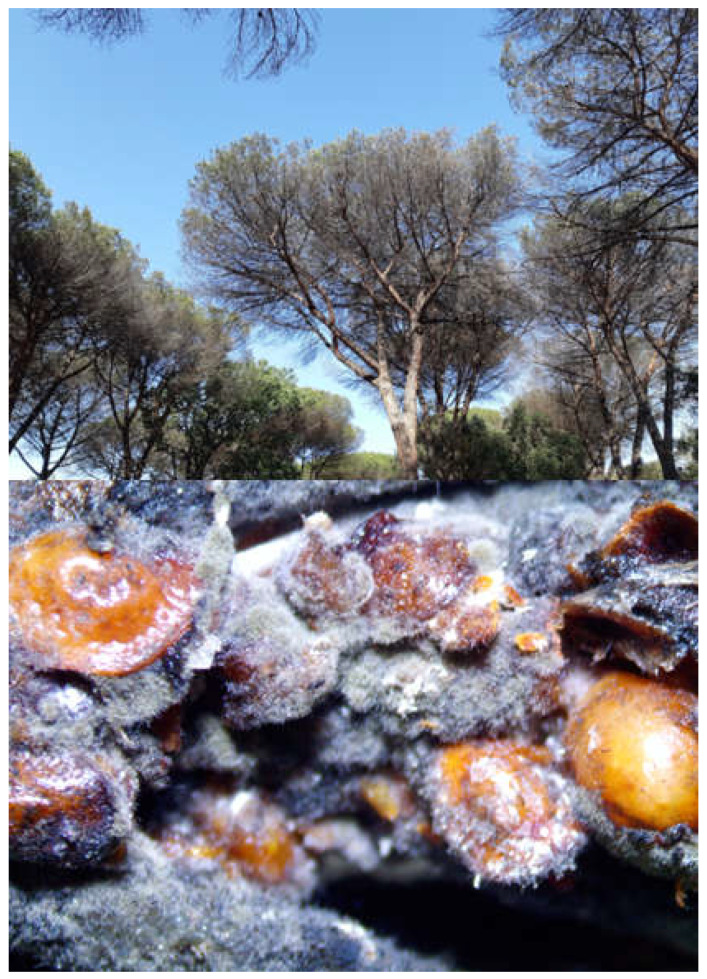
Symptoms on *Pinus pinea* plants infested by *Toumeyella parvicornis*: dying tree (**up**) and infested twig with adult females, honeydew, and mold (**bottom**).

**Figure 3 insects-14-00046-f003:**
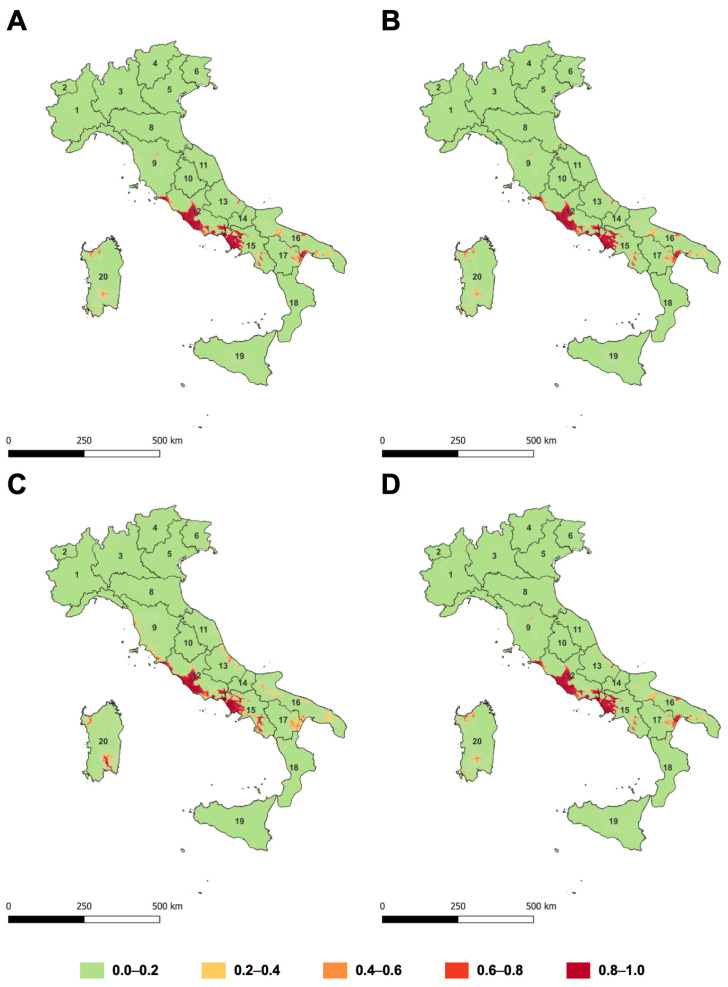
Distribution of suitable areas for *Toumeyella parvicornis* in Italy based on bioclimatic variables using Generalized Linear Model (GLM) (**A**), Multivariate Adaptive Regression Splines (MARS) model (**B**), MaxEnt model (**C**), and Random Forest (RF) model (**D**). Different colours indicate “classes of suitability” (green = very low; yellow = low; light orange = intermediate; dark orange = high; red = very high). Numbers in maps refer to Italian administrative regions (1 = Piemonte; 2 = Valle d’Aosta; 3 = Lombardia; 4 = Trentino Alto Adige; 5 = Veneto; 6 = Friuli Venezia Giulia; 7 = Liguria; 8 = Emilia Romagna; 9 = Toscana; 10 = Umbria; 11 = Marche; 12 = Lazio; 13 = Abruzzo; 14 = Molise; 15 = Campania; 16 = Puglia; 17 = Basilicata; 18 = Calabria; 19 = Sicilia; 20 = Sardegna).

**Figure 4 insects-14-00046-f004:**
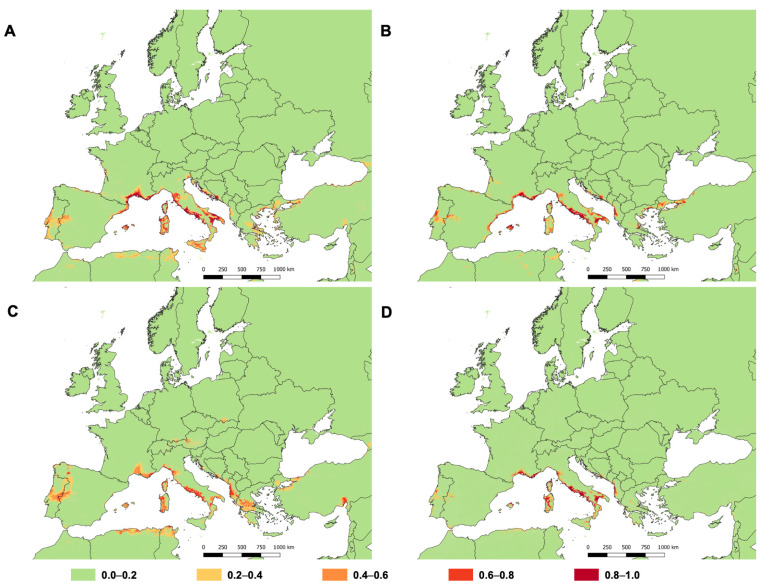
Distribution of suitable areas for *Toumeyella parvicornis* in Mediterranean Europe based on bioclimatic variables using Generalized Linear Model (GLM) (**A**), Multivariate Adaptive Regression Splines (MARS) model (**B**), MaxEnt model (**C**), and Random Forest (RF) model (**D**). Different colours indicate “classes of suitability” (green = very low; yellow = low; light orange = intermediate; dark orange = high; red = very high).

**Figure 5 insects-14-00046-f005:**
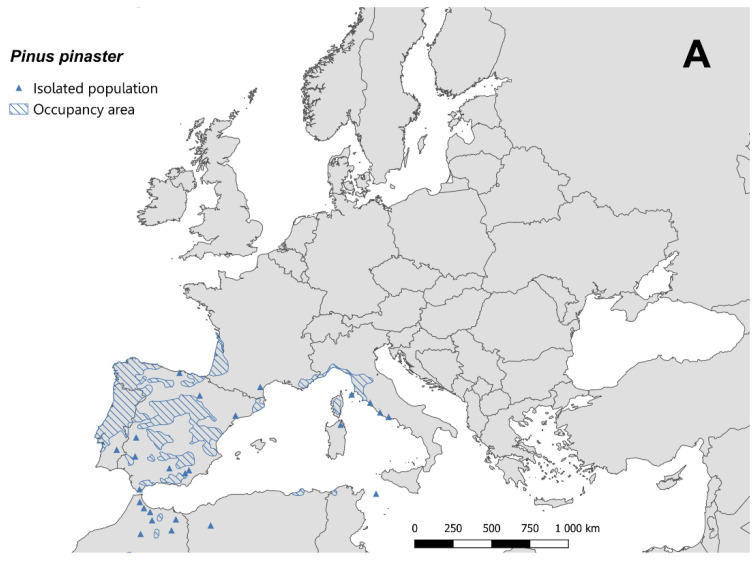
Distribution of (**A**) *Pinus pinaster* and (**B**) *Pinus pinea* in Europe following Caudullo et al. [79].

**Table 1 insects-14-00046-t001:** List of bioclimatic variables considered for modelling. Variables used in the Species Distribution Models (SDMs) are reported in bold.

Variable	Description
**BIO01**	**Annual Mean Temperature**
**BIO02**	**Mean Diurnal Temperature Range [Mean of monthly (max temp–min temp)]**
**BIO03**	**Isothermality (BIO02/BIO07) (×100)**
**BIO04**	**Temperature Seasonality (standard deviation × 100)**
BIO05	Max Temperature of Warmest Month
BIO06	Min Temperature of Coldest Month
BIO07	Temperature Annual Range (BIO05–BIO06)
**BIO08**	**Mean Temperature of Wettest Quarter**
**BIO09**	**Mean Temperature of Driest Quarter**
BIO10	Mean Temperature of Warmest Quarter
BIO11	Mean Temperature of Coldest Quarter
**BIO12**	**Annual Precipitation**
BIO13	Precipitation of Wettest Month
BIO14	Precipitation of Driest Month
**BIO15**	**Precipitation Seasonality (Coefficient of Variation)**
BIO16	Precipitation of Wettest Quarter
BIO17	Precipitation of Driest Quarter
BIO18	Precipitation of Warmest Quarter
**BIO19**	**Precipitation of Coldest Quarter**

**Table 2 insects-14-00046-t002:** Relative contribution of each bioclimatic variable to Generalized Linear Model (GLM), Multivariate Adaptive Regression Splines (MARS) model, MaxEnt model, and Random Forest (RF) model to estimate the suitability of Italy for *Toumeyella parvicornis*.

Bioclimatic Variable ^1^	GLM	MARS	MaxEnt	RF
BIO01	**19.74%**	**32.30%**	**33.12%**	**26.47%**
BIO02	6.40%	4.61%	3.07%	1.76%
BIO03	10.18%	**20.37%**	14.17%	**32.35%**
BIO04	6.05%	1.23%	5.14%	1.47%
BIO08	12.96%	2.65%	7.36%	5.59%
BIO09	5.77%	11.54%	3.66%	5.29%
BIO12	**16.12%**	2.21%	12.55%	7.06%
BIO15	12.18%	17.92%	**16.56%**	10.88%
BIO19	10.60%	7.17%	4.38%	9.12%

^1^ See Table 1 for acronyms.

**Table 3 insects-14-00046-t003:** Relative contribution of each bioclimatic variable to Generalized Linear Model (GLM), Multivariate Adaptive Regression Splines (MARS) model, MaxEnt model, and Random Forest (RF) model to estimate the suitability of Mediterranean Europe for *Toumeyella parvicornis*.

Bioclimatic Variable ^1^	GLM	MARS	MaxEnt	RF
BIO01	**35.73%**	**35.13%**	2.88%	**15.64%**
BIO02	8.44%	14.39%	10.20%	1.82%
BIO03	3.49%	2.86%	14.94%	4.73%
BIO04	6.36%	1.77%	2.64%	15.27%
BIO08	3.54%	0.64%	11.28%	11.27%
BIO09	6.49%	11.62%	6.66%	12.73%
BIO12	11.36%	2.27%	**15.42%**	12.00%
BIO15	4.49%	12.30%	**20.76%**	9.09%
BIO19	**20.09%**	**19.02%**	15.24%	**17.45%**

^1^ See Table 1 for acronyms.

## Data Availability

The data associated with this publication, as well as the R script to reproduce the results, are publicly available at https://github.com/lucaros1190/SDM-Toumeyella (accessed on 1 December 2022).

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
