# Peer review of "Using Species Distribution Models (SDMs) to Estimate the Suitability of European Mediterranean Non-Native Area for the Establishment of Toumeyella Parvicornis (Hemiptera: Coccidae)"

_insects, 2023, doi:10.3390/insects14010046_

Round 1

Reviewer 1 Report

 This study will be a useful contribution for researchers and land managers concerned with the invasive pine tortoise scale. The use of multiple models convincingly demonstrates their concurrence in showing regions of concern for potential range expansion. In general, the manuscript is well written and clear, although in scattered places the phrasing, use of grammar and style are not quite right for English fluency. In those places I have suggested alternative text (by line numbers, below). Most of my edits are relatively minor, and my overall recommendation is for publication with only minor revisions. Note also my suggestion to include Supplemental figure 3 as a figure in the main article.

Line #:

18          insert “is” between “species” and “a soft scale”

23          should read “seem suitable” to agree grammatically with previous subjects (plural)

23-30   There are numerous minor grammatical errors in these lines. I suggest rewriting these lines as follows [also note – the acronym EFSA should be spelled out in full]: “To prevent further spread across the Mediterranean basin it would be helpful to identify the most suitable areas by considering bioclimatic variables, as is commonly carried out in case of invasive species. We prepared potential pest distribution maps of European areas by utilizing Species Distribution Models. This information adds further detail to the report recently published by EFSA. The areas with the highest suitability for the species are located along the coasts, where most Mediterranean pines occur. This correspondence suggests a high risk of widespread dispersal and provides useful information for implementing management strategies of this damaging pest.”

33          “level” should be “levels”

36          suggest replacing the first “using” in this sentence with “by constructing” (closely following repetition of the same word in a sentence is awkward style)

41-43   suggest rewriting this as “Predicting the suitability of European Mediterranean areas for T. parvicornis provides a fundamental tool for early detection and management of the spread of this pest in Europe.”

56-57   rephrase “where it has been firstly described in Florida” as “where it was first described in Florida”

68          “increases” should be “increase”

70          rephrase “which may generate disturbance to urban park users” as “which can disturb urban park users”

 Method & Materials – Not being a modeler, I cannot comment in detail on the technical accuracy of the models that are described. However, this section seems clearly presented and well-written.

126-127   “raster images were clipped using … to assumed extent” – the meaning of this in not clear & needs to be explained further.

183-184  Rewrite this as “The variables with the greatest contributions to the distribution of T. parvicornis…”

203-204  rewrite as “The potential T. parvicornis distribution area was mainly concentrated…”

205       insert “the” before “Sardinia region”

210-211  Rewrite: “On the contrary, the potential distribution estimated by the RF model was quite restricted.”

223       replace “noticed” with ‘noted”

224       replace "to” with “for”

245       Riviera should be capitalized (as French Riviera is a specific place name)

246, 248   “North-Western” and “Eastern” should not be capitalized (as they are descriptive adjectives, and not part of an official place name)

259       delete “that”

263       replace “providing” with “estimated” or “predicted”

275       rearrange “variable explaining the best…” to “variable best explaining…”

283-284   rephrase “as Italian maritime areas normally are,” with “which Italian maritime areas normally have,”

289       replace “showed more extended” with “provided more extensive”

301       Replace “Differently,” with “In contrast,”

References – nearly four full pages of reference citations seems excessive for a 15-page article. Are these all really necessary?

Supplementary Fig. 3 – Why is this figure listed as Supplemental? In my opinion, this figure should instead be included as part of the main article and not in a supplement, as the distribution of these pines are a core aspect of the concern over the potential scale distribution.

Author Response

Response to Reviewer 1

Reviewer 1: This study will be a useful contribution for researchers and land managers concerned with the invasive pine tortoise scale. The use of multiple models convincingly demonstrates their concurrence in showing regions of concern for potential range expansion. In general, the manuscript is well written and clear, although in scattered places the phrasing, use of grammar and style are not quite right for English fluency. In those places I have suggested alternative text (by line numbers, below). Most of my edits are relatively minor, and my overall recommendation is for publication with only minor revisions. Note also my suggestion to include Supplemental figure 3 as a figure in the main article.

Response: Dear Reviewer 1, thank you very much for the time dedicated to revising our manuscript and for the constructive comments and suggestions as well. We have sincerely appreciated the positive comments about our manuscript, and we are grateful for this consideration of our work. Additionally, we wish to thank you for the detailed comments about English style, that for non-native speakers is always a source of learning. We renew our availability for any further requests or changes if needed, and we hope that this revised version fits your expectations. Thank you again.

Reviewer 1: Line #18: insert “is” between “species” and “a soft scale”

Response: Thank you for this suggestion, we have corrected the text accordingly.

Reviewer 1: Line #23: should read “seem suitable” to agree grammatically with previous subjects (plural)

Response: Thank you for this suggestion, we have corrected the text accordingly.

Reviewer 1: Lines #23-30: There are numerous minor grammatical errors in these lines. I suggest rewriting these lines as follows [also note – the acronym EFSA should be spelled out in full]: “To prevent further spread across the Mediterranean basin it would be helpful to identify the most suitable areas by considering bioclimatic variables, as is commonly carried out in case of invasive species. We prepared potential pest distribution maps of European areas by utilizing Species Distribution Models. This information adds further detail to the report recently published by EFSA. The areas with the highest suitability for the species are located along the coasts, where most Mediterranean pines occur. This correspondence suggests a high risk of widespread dispersal and provides useful information for implementing management strategies of this damaging pest.”

Response: Thank you for this suggestion, we have corrected the text accordingly.

Reviewer 1: Line #33: “level” should be “levels”

Response: Thank you for this suggestion, we have corrected the text accordingly.

Reviewer 1: Line #36: suggest replacing the first “using” in this sentence with “by constructing” (closely following repetition of the same word in a sentence is awkward style)

Response: Thank you for this comment and also for pointing out why the suggested word is more suitable! We have corrected the text accordingly.

Reviewer 1: Lines #41-43: suggest rewriting this as “Predicting the suitability of European Mediterranean areas for T. parvicornis provides a fundamental tool for early detection and management of the spread of this pest in Europe.”

Response: Thank you for this suggestion, we have corrected the text accordingly.

Reviewer 1: Lines #56-57: rephrase “where it has been firstly described in Florida” as “where it was first described in Florida”

Response: Thank you for this suggestion, we have corrected the text accordingly.

Reviewer 1: Line #68: “increases” should be “increase”

Response: Thank you for this suggestion, we have corrected the text accordingly.

Reviewer 1: Line #70: rephrase “which may generate disturbance to urban park users” as “which can disturb urban park users”

Response: Thank you for this suggestion, we have corrected the text accordingly.

Reviewer 1: Method & Materials – Not being a modeler, I cannot comment in detail on the technical accuracy of the models that are described. However, this section seems clearly presented and well-written.

Response: Thank you very much for this positive comment, that we have sincerely appreciated.

Reviewer 1: Lines #126-127: “raster images were clipped using … to assumed extent” – the meaning of this in not clear & needs to be explained further.

Response: Thank you for this comment, we have rephrased this part of the text accordingly.

Reviewer 1: Lines #183-184: Rewrite this as “The variables with the greatest contributions to the distribution of T. parvicornis…”

Response: Thank you for this suggestion, we have corrected the text accordingly.

Reviewer 1: Lines #203-204: rewrite as “The potential T. parvicornis distribution area was mainly concentrated…”

Response: Thank you for this suggestion, we have corrected the text accordingly.

Reviewer 1: Line #205: insert “the” before “Sardinia region”

Response: Thank you for this suggestion, we have corrected the text accordingly.

Reviewer 1: Lines #210-211: Rewrite: “On the contrary, the potential distribution estimated by the RF model was quite restricted.”

Response: Thank you for this suggestion, we have corrected the text accordingly.

Reviewer 1: Line #223: replace “noticed” with ‘noted”

Response: Thank you for this suggestion, we have corrected the text accordingly.

Reviewer 1: Line #224: replace "to” with “for”

Response: Thank you for this suggestion, we have corrected the text accordingly.

Reviewer 1: Line #245: Riviera should be capitalized (as French Riviera is a specific place name)

Response: Thank you for this suggestion, we have corrected the text accordingly.

Reviewer 1: Lines #246, 248: “North-Western” and “Eastern” should not be capitalized (as they are descriptive adjectives, and not part of an official place name)

Response: Thank you for this suggestion, we have corrected the text accordingly.

Reviewer 1: Line #259: delete “that”

Response: Thank you for this suggestion, we have corrected the text accordingly.

Reviewer 1: Line #263: replace “providing” with “estimated” or “predicted”

Response: Thank you for this suggestion, we have corrected the text accordingly.

Reviewer 1: Line #275: rearrange “variable explaining the best…” to “variable best explaining…”

Response: Thank you for this suggestion, we have corrected the text accordingly.

Reviewer 1: Lines #283-284: rephrase “as Italian maritime areas normally are,” with “which Italian maritime areas normally have,”

Response: Thank you for this suggestion, we have corrected the text accordingly.

Reviewer 1: Line #289: replace “showed more extended” with “provided more extensive”

Response: Thank you for this suggestion, we have corrected the text accordingly.

Reviewer 1: Line #301: Replace “Differently,” with “In contrast,”

Response: Thank you for this suggestion, we have corrected the text accordingly.

Reviewer 1: References – nearly four full pages of reference citations seems excessive for a 15-page article. Are these all really necessary?

Response: Thank you very much for this comment. We have carefully evaluated the suggestion to reduce the amount of references, but are all essential to understanding the biology of the species or the choices and assumptions behind the models used. We hope that this choice fits with your expectations, and that the potential readers can benefit from having in a single manuscript all the information to better explore both the biology of the species and the modelling part.

Reviewer 1: Supplementary Fig. 3 – Why is this figure listed as Supplemental? In my opinion, this figure should instead be included as part of the main article and not in a supplement, as the distribution of these pines are a core aspect of the concern over the potential scale distribution.

Response: Thank you very much for this suggestion. When we were preparing the initial draft of the manuscript, we thought about inserting the pine distribution map in the main text. We have decided to move it into the supplementary files because we had the impression that it was quite redundant with the other maps. Your comment, however, confirmed our initial idea and we have moved this figure in the main text accordingly.

Thank you again for your kind support, we have sincerely appreciated the professionality and the quality of your revision!

Reviewer 2 Report

This study investigated the species distribtion models (SDMs) to estimate the implementation of Toumeyella parvicornis. It a very nice study and well written.

Title: You could add "," or ":" between "Hemiptera" and "Coccidae"

Intro: Maybe a picture of the adult and a schema of the cycle could be very instructive for the readership.

69-71: in what case the honeydew produced could cause disturbance to urban park users ?

The rest of the introduction is very clear and well written. But I wonder what are the different actions that could control the targeted pest ?

113-116: is it possible to have a picture of the typical symptoms ?

129-130: The choice to to remove variable showing R2 > 0.80 or R2 < -0.80 is arbitrary ? Is it based on literature ? Can you justify it ?

Table 1:  Why variables in BOLD are kept for the SDMs analysis ?

134 : Why do you use four different algorithms and not one ?

Table 2 : Explain why some proportions are in bold ? Same for Table 3

Figure 2: Is it possible to increase the resolution and the size of the maps ?

The discussion is fine for me 

Author Response

Response to Reviewer 2

Reviewer 2: This study investigated the species distribution models (SDMs) to estimate the implementation of Toumeyella parvicornis. It is a very nice study and well written.

Response: Dear Reviewer 2, thank you very much for the time dedicated to revising our manuscript and for the constructive comments and suggestions as well. We have sincerely appreciated the positive comments about our manuscript, and we are grateful for this consideration of our work. Additionally, we wish to thank you for the detailed comments and questions about some minor aspects of the manuscript. We renew our availability for any further requests or changes if needed, and we hope that this revised version fits your expectations. Thank you again.

Reviewer 2: Title: You could add "," or ":" between "Hemiptera" and "Coccidae"

Response: Thank you very much for pointing out this misprint. We have corrected the title accordingly, inserting the missing “:”.

Reviewer 2: Intro: Maybe a picture of the adult and a schema of the cycle could be very instructive for the readership.

Response: Thank you very much for this comment. Part of this response is related to the comment below, concerning pictures of the damage. We have sincerely appreciated the suggestion of inserting both an image of the insect and of the damage caused. We hope that our corrections fits with your expectations.

Reviewer 2: 69-71: in what case the honeydew produced could cause disturbance to urban park users?

Response: Thank you for this question. We have better clarified in the text which is the disturbance caused by the honeydew in urban parks. We hope that this part of the manuscript better fits with your expectations.

Reviewer 2: The rest of the introduction is very clear and well written. But I wonder what are the different actions that could control the targeted pest?

Response: Thank you for this question. The control of this pest is not completely explored, and we have several ongoing experimentations to better explore this aspect. To date, only the endotherapic abamectin has been applied to stone pine plants, but their scarce persistence requires periodic treatments. We decided to not include too much of this information in this manuscript, to not divert the attention of the reader from the main aim. On the other hand, the references were selected to complete the set of information that in this context may generate confusion in understanding of the aim of our work.

Reviewer 2: 113-116: is it possible to have a picture of the typical symptoms ?

Response: Thank you very much for this comment. As told in a few responses above, we have inserted a picture of the damage and of the insect, so that the manuscript may be more clear to potential readers. We hope that this change fits with your expectations.

Reviewer 2: 129-130: The choice to remove variable showing R2 > 0.80 or R2 < -0.80 is arbitrary ? Is it based on literature? Can you justify it?

Response:  Thank you for this comment. The choice of removing variables belonging to the above-mentioned range is based on the methodologies commonly described in the literature. All the supporting references are already listed in the bibliography, so that the most interested readers can better explore this aspect. The use of SDMs makes it necessary to run a correlation analysis between the variables, in order to avoid possible reductions of the quality of the models’ outputs.

Reviewer 2: Table 1:  Why variables in BOLD are kept for the SDMs analysis ?

Response: Thank you for this comment. The variables in bold were considered for the SDMs analysis because they did not result, after the multicollinearity analysis, as highly correlated. To avoid confusion in potential readers we have better explained this part in the text.

Reviewer 2: 134 : Why do you use four different algorithms and not one?

Response: Thank you very much for this question. There are different points of view about this aspect among the scientific community. After a deep literature review, we have noticed that some authors simply choose one among the models available in the literature, motivating the choice, while other authors prefer to compare outputs from different models to analyse where they are more coherent. We preferred this second approach, since it is hard to choose a priori a model and entrust to it the potential distribution of a species. Given the scarcity of biological information and data about this species, we have preferred to explore and compare outputs from different models to maximize the information to discuss.

Reviewer 2: Table 2 : Explain why some proportions are in bold ? Same for Table 3

Response: Thank you for this comment. Table 2 and Table 3 report in bold only the highest values of the obtained proportions, in order to make more comfortable the reading of the results.

Reviewer 2: Figure 2: Is it possible to increase the resolution and the size of the maps ?

Response: Thank you very much for this comment. We have provided the electronic figures as separate files, so that at least in the online version of the manuscript it will be possible to zoom the points of interest. In the manuscript, instead, it is difficult to enlarge even more the figures, that are already at the limit of the size.

Reviewer 2: The discussion is fine for me

Response: Thank you very much for this positive comment. We have sincerely appreciated your support, the professionality and the quality of your review!